# Comprehensive Volatilome and Metabolome Signatures of Colorectal Cancer in Urine: A Systematic Review and Meta-Analysis

**DOI:** 10.3390/cancers13112534

**Published:** 2021-05-21

**Authors:** Celia Mallafré-Muro, Maria Llambrich, Raquel Cumeras, Antonio Pardo, Jesús Brezmes, Santiago Marco, Josep Gumà

**Affiliations:** 1Department of Electronics and Biomedical Engineering, University of Barcelona, 08028 Barcelona, Spain; cmallafre@ibecbarcelona.eu (C.M.-M.); a.pardo@ub.edu (A.P.); smarco@ibecbarcelona.eu (S.M.); 2Signal and Information Processing for Sensing Systems Group, Institute for Bioengineering of Catalonia, The Barcelona Institute of Science and Technology, 08028 Barcelona, Spain; 3Metabolomics Interdisciplinary Group (MiL@b), Department of Electrical Electronic Engineering and Automation, Universitat Rovira i Virgili (URV), IISPV, CERCA, 43007 Tarragona, Spain; maria.llambrich@urv.cat (M.L.); jesus.brezmes@urv.cat (J.B.); 4Biomedical Research Centre, Diabetes and Associated Metabolic Disorders (CIBERDEM), ISCIII, 28029 Madrid, Spain; 5Fiehn Lab, NIH West Coast Metabolomics Center, University of California Davis, Davis, CA 95616, USA; 6Oncology Department, Hospital Universitari Sant Joan de Reus, Institut d’Investigació Sanitària Pere Virgili (IISPV), Universitat Rovira i Virgili (URV), 43204 Reus, Spain; jguma@grupsagessa.com

**Keywords:** colorectal cancer, metabolomics, volatilomics, systematic review, meta-analysis, urine

## Abstract

**Simple Summary:**

Colorectal cancer is the most frequent neoplasm in Western countries, the second most frequent neoplasm after breast cancer in women and the third most frequent neoplasm after prostate and lung cancer in men. Early diagnosis and disease screening are based on a fecal occult blood test, which, if positive, is complemented by colonoscopy. Currently, efforts are underway to find alternatives to the fecal occult blood test for various reasons. First, there is an ongoing attempt to increase the participation of the population to be screened. Second, there is a need to decrease the number of false positives to reduce the number of unnecessary colonoscopies. A urine test could be more widely accepted than a fecal test, and this is the scenario for which urinary metabolomics and volatilome studies are being developed. Our review provides the first exhaustive evaluation of metabolomics and volatilomics for the determination of colorectal cancer in urine.

**Abstract:**

To increase compliance with colorectal cancer screening programs and to reduce the recommended screening age, cheaper and easy non-invasiveness alternatives to the fecal immunochemical test should be provided. Following the PRISMA procedure of studies that evaluated the metabolome and volatilome signatures of colorectal cancer in human urine samples, an exhaustive search in PubMed, Web of Science, and Scopus found 28 studies that met the required criteria. There were no restrictions on the query for the type of study, leading to not only colorectal cancer samples versus control comparison but also polyps versus control and prospective studies of surgical effects, CRC staging and comparisons of CRC with other cancers. With this systematic review, we identified up to 244 compounds in urine samples (3 shared compounds between the volatilome and metabolome), and 10 of them were relevant in more than three articles. In the meta-analysis, nine studies met the criteria for inclusion, and the results combining the case-control and the pre-/post-surgery groups, eleven compounds were found to be relevant. Four upregulated metabolites were identified, 3-hydroxybutyric acid, L-dopa, L-histidinol, and N1, N12-diacetylspermine and seven downregulated compounds were identified, pyruvic acid, hydroquinone, tartaric acid, and hippuric acid as metabolites and butyraldehyde, ether, and 1,1,6-trimethyl-1,2-dihydronaphthalene as volatiles.

## 1. Introduction

There is epidemiological importance of colorectal cancer (CRC) in developed countries, as it is the most common tumor when both sexes are considered together. In these geographic areas, it is the third leading cause of cancer mortality in men and the second in women [1]. The incidence of CRC increases with age, with most of the cases diagnosed over 50 years of age. Regarding the risk factor, 90% of CRC cases are considered sporadic (non-hereditary), and the rest may be associated with familial cancer syndromes such as familial adenomatous polyposis or Lynch syndrome (hereditary nonpolyposis colorectal cancer) [2]. Another risk factor is a diet rich in red or processed meat, however, fiber, vegetables, and fruit are protective [3,4]. All these dietary factors modify the risk of colorectal adenomas, the premalignant lesion of CRC. Obesity is another risk factor, and exercise and physical activity act as protectors [5]. Therefore, CRC is considered to be caused by a combination of genetic and environmental factors that lead to the appearance of adenomatous polyps as premalignant lesions, which acquire new genetic mutations over time until cancer occurs. The most frequent symptoms of colorectal cancer are changes in bowel habits and the appearance of blood in the stool. The main diagnostic tool is fibrocolonoscopy, which allows for tumor detection and biopsy collection in the same procedure [6]. Currently, the detection of occult blood in feces (FOBT) is the most widely used diagnostic test for screening for colorectal cancer in asymptomatic patients. This test is recommended for healthy individuals between 50 and 70 years of age, with biennial periodicity, and it can reduce mortality from CRC by approximately 25% [7]. FOBT can be implemented with different modalities, namely, guaiac or fecal immunochemical testing (FIT). While it has been reported that the second outperforms the first [8], FIT shows large variability in its sensitivity and positive predictive value (PPV). Despite the improvements in the performance of fecal tests, the number of false positives (FP) largely exceeds the number of true positives (TP). The performance of these tests depends on a number of technical details, such as the number of samples collected from each patient, the hemoglobin threshold, and the specific kit used. PPV also depends on the patient age, with lower PPVs observed in younger patients. Other factors affecting the PPV are sex (lower for females) and colorectal condition (lower for patients with a previous colorectal clinical history) [9]. Diverse studies report PPV values in the range of 10–30%, while the area under the ROC curve (AUC) is typically in the range of 0.7–0.9 [10,11]. Improvements in CRC screening are needed to minimize the cost and potential complications of subsequent colonoscopy.

The metabolome is the complete set of small molecules found in a biological sample. For this study, we focused on metabolites present in biological fluids from the human body, such as blood, urine, sweat, or saliva. The volatilome is the volatile fraction of the metabolome, and it produces aroma signatures for certain diseases that have been known to medical practice since Hippocrates [12,13]. Volatilome studies originally focused on breath samples, but they have expanded into biofluids [14]. Alterations in METs and VOLs can provide valuable information about the normal or abnormal performance of the body (due to disease) at a given time.

Urine is available in larger quantities than blood, is obtained in a noninvasive manner, and is mostly free from interfering proteins or lipids. However, urine is quite chemically complex because it is not only a biofluid of clinical importance but also a waste breakdown product of foods and beverages, drugs, environmental contaminants, endogenous waste metabolites, and bacterial byproducts [15]. Currently, up to ~3100 small molecules in human urine have already been identified (https://urinemetabolome.ca/ (accessed on 10 November 2020)).

In this study, with a special focus on colorectal cancer, we performed a systematic review and meta-analysis of the metabolome and volatilome to provide a comprehensive clinical significance of possible metabolomics and volatilomics-based biomarkers from urine samples. During the systematic review process, we included articles from adenomas or polyps, as they are precursors of colorectal cancer. Additionally, a meta-analysis of the study results was conducted on the basis of the ethical storage and quality of the urine samples obtained, the use of a minimum statistically significant number of samples per group, and the identification of the reported compounds.

## 2. Materials and Methods

### 2.1. The PRISMA Method

Bibliography research and article selection were performed using the Preferred Reporting Items for Systematic Reviews and Meta-Analyses or PRISMA method [16], which is the most common method used for systematic reviews. A double-screening approach was employed [17]. Two researchers independently performed both the search sentence and all the data evaluation and selection, and any discrepancy was resolved by discussion.

### 2.2. Search Sentence

The search was done using a search sentence on 14 July 2020, on the repositories: PubMed (https://pubmed.ncbi.nlm.nih.gov/), Web of Science (WOS) (https://webofknowledge.com/), and SCOPUS (https://www.scopus.com/). The search sentence was constructed as follows TITLE-ABS-KEY ((urine OR urinary OR urinate OR urination) AND (colorectal OR colon) AND (tumor OR tumour OR malignancy OR neoplasm OR cancer OR carcinoma OR adenoma OR polyps OR polyp) AND (human OR humans) AND (volatile OR volatiles OR {metabolite profiling} OR {metabolite analysis} OR {metabolic profiling} OR {metabolic fingerprinting} OR {metabolic characterization} OR metabolome OR metabolomics OR metabolomic OR metabonomics OR metabonomic OR lipidome OR lipidomics OR lipidomic)).

### 2.3. Inclusion and Exclusion Criteria

Data obtained on the search were combined in an Excel file including title, year of publication, authors, and abstracts. First, by reading the title and the abstract we removed the duplicated articles. Only the research articles were included, and other types of publications were removed such as reviews, book chapters, or conference papers. Further evaluation of the articles for eligibility was made reading the full text, and in this step at least two authors reviewed it to avoid biases. If there were inconsistencies, the decision was made by consensus. Other parameters for study exclusion used were, if matrix was not urine, studies conducted on animals or cell lines, was not colorectal cancer, or was related to food or drug outcomes. However, we included all types of study design, race, geographical area, or population for the systematic review. For meta-analysis extra restriction was considered, the study design had to fulfill the following: (A) minimum *N* = 20 per group to ensure statistical significance [18]; (B) study groups were matched by age and sex; (C) compounds must be identified and behavior reported (up/down); (D) ethics approval must be reported; and (E) urine storage conditions must be reported, since urine compounds degrade in long-term storage if temperature is higher than −20 °C [19].

### 2.4. Meta-Analysis

Due to the limited data availability, a meta-analysis was conducted with the statistical information reported in the studies. For each compound, we collected the *p*-value and fold-change. *p*-values were combined using the Fisher test, which combines the squares of the *p*-value, considering that *p*-values should be uniformly distributed, and compares them to a chi squared distribution, and weighting by the number of individuals, as described elsewhere [20,21]. Weighted *p*-values include the estimations in accordance with the number of individuals in the study. Depending on the number of individuals in the study, the results will be more trustworthy (an elevated number of individuals in the study gives more confidence in the results). Fold-change was logarithmically transformed and averaged with weighting by the number of individuals in the study. To maintain the trends of the results, records were divided into upregulated and downregulated compounds.

## 3. Results

The results were divided into five parts: (1) PRISMA process results; (2) characteristics of the included studios for both qualitative (systematic review) and quantitative (meta-analysis) results; (3) systematic review results; (4) quality assurance results for the studies included; and (5) meta-analysis results.

### 3.1. PRISMA Process

The whole process of the PRISMA method is shown in Figure 1. The search returned a total of 220 reports from Scopus (112), Web of Science (54), and PubMed (54), plus 7 additional records that were identified through other sources. From these, up to 161 studies were included for title and abstract screening after deleting duplications. We then excluded 51 studies that were not related to the study question or were reviews, conference papers, book chapters, short surveys, notes, letters, or editorials. This yielded a total of 110 studies eligible for further full-text assessment. We excluded 82 publications because the matrix did not fit the query (no urine), the participants were not human (mainly mice), the specimens were not colorectal cancer samples or were drug- or food-related, or the articles were reviews. The final list of included studies for the systematic review (qualitative synthesis) contained 28 papers, and from those, only 9 were included for quantitative synthesis in the meta-analysis, which met the criteria of minimum compounds per group and matched groups by age and sex, compound identification, ethics approval, and urine sample storage.

### 3.2. Characteristics of the Included Studies

For the 28 studies meeting the criteria on systematic review, namely, 7 for volatiles and 21 for metabolites, we prepared comprehensive tables divided on the methodology of the study (Table 1), cohort information (Table 2), identified compounds (Appendix A), and nonidentified but reported compounds (Appendix A). The 28 studies included can be classified as 22 CRC/control studies (colorectal cancer samples, of which 7 also include other cancer types for study), 3 investigating adenoma samples, and 3 analyzing both colorectal cancer and adenoma samples (see Table 1). Different methodology strategies were used. The most common was case-control analysis (24 studies), followed by the evaluation of samples before and after tumor extraction (3 studies), and in 1 study [22], a time prospective study was conducted, taking into account relapse (occurrence of tumors after removal). The principal objective of the cited research was to identify compounds characteristic of colorectal cancer; 4 studies used a targeted approach [23,24,25,26], and the rest performed untargeted research. However, only 2 articles externally validated [23,27] the results obtained. More commonly, internal validation was performed; however, less than 50% of studies disclosed this. In all articles, except for four studies that did not mention information, samples were always stored at −80 °C to avoid sample degradation. The techniques used to analyze samples were diverse, with most of the articles using common approaches such as nuclear magnetic resonance (NMR), liquid chromatography-mass spectrometry (LC-MS), and gas chromatography (GC-MS); in some studies, they also tested new devices such as field asymmetric waveform ion mobility spectrometry (FAIMS) [28], needle trap microextraction [29] or E-nose [30] for gases. Finally, capillary electrophoresis (CE-MS) [31] and rapid-resolution liquid chromatography–time-of-flight mass spectrometry (RRLC-TOF/MS) [32] was employed for liquids. Urine collection also differed between studies. One study collected first morning urine after fasting to avoid interferences from food or lifestyle in the samples. However, in 14 cases, spot urine was used, or information was not disclosed about the methodology followed.

The tables obtained included 29 cohorts reported in 28 studies (Table 2). There was only one study [24] that used two different cohorts. Nevertheless, descriptions of cohort information were only complete in 12 studies, considering complete information descriptions of participants should include age and stages of cancer. Additional information (Appendix A) about smoking history and body mass index (BMI) was presented in 11 reports, but only 3 reports provided both types of information [28,30,33]. Alcohol consumption was disclosed in 2 studies from the VOC phase due to its importance to the results [28,30]. In total, researchers from 11 countries have studied compounds from urine, and all these countries have a high CRC incidence and mortality rate (Appendix A). The country with the most studies is China, with 12 of the selected ones, followed by the United Kingdom, with 5 studies. All the countries with included studies have a colorectal screening program in place. Currently, only 40 countries worldwide have a running screening program [34]. The increase in incidence and mortality in countries with a high population can increase health system costs, prompting further investigation. Fewer than 100 participants were enrolled in 8 studies (Appendix A), with only 34 participants in total in the smallest study [31].

**Table 1 cancers-13-02534-t001:** Methodological information from the systematic review (qualitative analysis) of selected studies. Asterisk * indicates that the study was used in the meta-analysis (quantitative analysis). Studies are ordered by type and then by reference.

Ref.	Kind	Platform	Type of Study	Ethics Approval	Urine Collection	Urine Storage	Analytical Validation	ROC Curve (Training/Testing)
[28]	VOL	FAIMS + GC-MS	CRC/control	yes	ND	−80 °C	1/2–1/2 repeated 5 times	-
* [35]	MET	GC + LC	CRC/control	yes	Fasting urine	−80 °C	2/3–1/3	0.993 (7 compounds)/0.998
[32]	MET	RRLC-TOF/MS	CRC/control	yes	First morning urine	−80 °C	2/3–1/3	-
* [36]	MET	NMR	CRC/control	yes	First morning urine	−80 °C	-	0.823 taurine, 0.783 alanine, 0.842 3-aminoisobutyrate/ND
[33]	MET	LC-FAIMS-MS	CRC/control	yes with ID	ND	−80 °C	-	0.71/ND
* [25]	MET	Targeted LC-MS/MS	CRC/control	yes with ID	Controls at 7–8 a.m., 11–12 a.m., and 5–6 p.m. CRC at 9 a.m. and 4 p.m.	−80 °C	bootstrapping with virtual datasets	0.794/ND
[31]	MET	CE-MS	CRC/control (including stages)	yes	Morning urine	−80 °C	-	0.906/ND
* [24]	MET	Targeted LC-MS/MS	CRC/control (including stages)	yes with ID	ND	−80 °C	yes	0.903/0.872
* [37]	MET	1H-NMR	CRC/control (including stages + other cancers)	yes	Fasting morning urine	−80 °C	80% training, 20% testing	0.875 alanine, 0.913 glutamine, 0.933 aspartic acid/ND
[38]	MET	HPLC-ESI-MS/MS	CRC/control (+ other cancers)	yes	ND	−80 °C	-	-
[39]	MET	LC-MS/MS MRM	CRC/control (+ other cancers)	yes	First morning urine	−80 °C	-	-
* [29]	VOL	Needle trap + GC-MS	CRC/control (+ other cancer)	yes	First morning urine	−80 °C	2/3–1/3	-
[40]	VOL	GC-MS	CRC/control (+ other cancers)	yes	ND	ND	-	-
[41]	VOL	GC-MS	CRC/control (+ other cancers)	yes	Fasting morning urine	−80 °C	-	-
[30]	VOL	E-nose	CRC/control (+ other diseases)	yes	Fasting morning urine	−80 °C	-	-
[42]	MET	RP-HPLC	CRC/control (along time)	yes	Spontaneous urine samples 1 day before surgery and day 8 after	−20 °C	-	0.896 1-methylguanosine, 0.816 pseudouridine/ND
[22]	MET	UPLC-QTOF-MS	CRC no-relapse/relapse	ND	ND	ND	2-fold cross-validation with 10,000 validations	AUC: 0.9675 (positive charge) and 0.95 (negative charge)/ND
* [43]	MET	GC-MS	CRC/control (pre-/post-surgery)	yes	Fasting morning urine	−80 °C	16/17–1/17	-
* [44]	MET	1H-NMR + GC-MS	CRC pre-/post-surgery and 6-/12-months follow-up AND intra-stages	yes	Pre-/post-surgery overnight fasting urine, 6-/12-months follow-up URINE spot	−80 °C	-	0.89 (20 compounds)/ND
[45]	MET	UPLC-MS	CRC/control (pre-/post-surgery + along time)	yes	Fasting urine (7:00 a.m.)	−80 °C	-	-
[26]	MET	targeted HPLC/GC-MS	CRC/adenoma/control	yes	Spot sample before surgery	ND	-	0.690 8-oxoGua, 0.635 8-oxoGuo, 0.669 5-hmUra/ND
[46]	MET	UPLC-MS/HPLC-MS	CRC/adenoma/control	yes	Morning fasting urine	−80 °C	7-fold	0.959 (12 compounds), 0.894 (7 nucleotides)/ND
[47]	MET	HPLC-MS/MS	CRC/adenoma/control	ND	Spot urine	−20 °C	-	-
[48]	VOL	FAIMS + GC-IMS	CRC/ adenoma/control (+ other diseases)	yes	ND	−80 °C	-	0.98/ND
[49]	VOL	FAIMS	CRC/adenoma/control (+ other cancers and diseases)	yes	Spot urine	−80 °C	-	0.9/ND
[23]	MET	NMR + targeted LC-MS/MS	Adenoma/control	yes with ID	Midstream urine	−80 °C ^ǂ^	2/3–1/3	0.687/0.692
[27]	MET	1D NMR	Adenoma/control	yes	Midstream urine	4 h at 4 °C 24 h at −80 °C	Validation of Deng L 2017	0.717/ND
* [50]	MET	1D NMR	Adenoma/control	yes	Midstream urine	4 h at 4 °C 24 h at −80 °C	2/3–1/3	0.752/ND

ND not disclosed. MS/MS tandem mass spectrometry, HPLC high-performance liquid chromatography, ESI electrospray ionization, MRM multiple reaction monitoring, RP reversed-phase, UPLC ultra-performance liquid chromatography, QTOF quadrupole time-of-flight. ^ǂ^ Urine temperature conditions reported in a previous publication.

**Table 2 cancers-13-02534-t002:** Cohort information from the systematic review (qualitative analysis) of selected studies. The asterisk * indicates that the study was used in the meta-analysis (quantitative analysis).

Ref. (Kind)	Group	*N*	Age (Error and Type)	Male/Female	Cancer Staging Classification (*n*)	Country
[28]	CRC	83	60 (ND: 17)	53/30	ND	UK
(VOL)	Control	50	47 (ND: 16)	21/29	-	
* [35]	CRC	101	60 (R: 24–83)	58/43	0 (0), I (24), II (45), III (27), IV (5)	CN
(MET)	Control	103	58 (R: 31–76)	31/72		
[32]	CRC	29	ND	-	ND	CN
(MET)	Control	10	ND	-	-	
* [36]	CRC	92	60 (R: 32–85)	62/30	0 (24), I (8), II (7), III (13), IV (4)	KR
(MET)	Control	156	52 (R: 22–76)	76/80		
[33]	CRC	56	65.4 (SD: 11.5)	33/23	A (8), B (17), C1 (20), C2 (9)	UK
(MET)	Control (spouse)	45	60.7 (SD: 12.1)	15/30	-	
	Control (relative)	37	50 (SD: 14.1)	17/20	-	
* [25]	CRC-Malignant	201	68.7 (ND: 0.8)	114/87	0 (3), I/II (103), III (88), IV (7)	JP
(MET)	CRC-Benign	14	65 (ND: 3.1)	11/3	-	
	Control	17	42.1 (ND: 2.8)	13/4	-	
[31]	CRC	20	73 (ND)	10/10	I/II (8), III/IV (12)	CN
(MET)	Control	14	68 (ND)	8/6	-	
* [24]	CRC-CAD	121	67.4 (ND: 10.9)	68/59	0 (3); I (16), II (30), III (51), IV (21)	CA/US
(MET)	CRC-MSKCC	50	63.8 (ND: 12.5)	24/26	0 (0), I (14), II (20), III (6), IV (10)	
	Control	171	58.9 (ND: 5.6)	100/71	-	
* [37]	CRC	55	60 (ND)	26/29	I/II (23), III/IV (32)	CN
(MET)	Control	40	59 (ND)	19/21	-	
	EC	18	61 (ND)	8/10	-	
[38]	CRC	26	65.3 (R: 33–88)	12/24	0 (0), I (3), II (6), III (10), IV (7)	TW
(MET)	Control	45	ND	-	-	
	LC	27	60.8 (R: 42–81)	16/11	-	
	GC	15	67.1 (R: 50–82)	12/3	-	
	BC	36	ND	-	-	
[39]	CRC	10	51.5 (SD: 6.6)	5/5	ND	CN
(MET)	Control	10	48.7 (SD: 6.43)	5/5	-	
	LC	10	52.5 (SD: 7.47)	5/5	-	
	NpC	10	49.3 (SD: 9.09)	5/5	-	
* [29]	CRC	30	ND (R: 45–83)	16/14	ND	PT
(VOL)	Control	30	ND (R: 18–78)	14/16	-	
	BC	30	ND (R: 38–83)	0/30	-	
[40]	CRC	8	ND	-	ND	ND
(VOL)	Control	35	ND	-	-	
	LC	14	ND	-	-	
	EC	12	ND	-	-	
	GC	12	ND	-	-	
[41]	CRC	11	62 (SD: 12.4 R: 49–78)	8/3	ND	PT
(VOL)	Control	21	62 (SD: 10.3 R: 28–60)	18/3	-	
	LeukC	14	50.1 (SD: 12.4 R: 40–74)	6/8	-	
	LyC	7	42 (SD: 19.1 R: 18–68)	6/1	-	
[30]	CRC	39	70 (ND)	28/11	ND	UK
(VOL)	Control	18	41 (ND)	13/5	-	
	IBS	35	48 (ND)	4/31	-	
[42]	CRC	52	63 (R: 26–87)	27/25	A (5), B (22), C (18), D (7)	CN
(MET)	Control	62	59 (R: 24–78)	33/29	-	
[22]	CRC non-relapse	20	ND	-	ND	
(MET)	CRC relapse	20	ND	-	ND	
* [43]	CRC	60	58.8 (ND)	34/26	0 (0), I (7), II (23), III (21), IV (9)	CN
(MET)	Control	63	55.5 (ND)	32/31	-	
* [44]	CRC pre-S	97	64.8 (SD: 12.9)	59/38	0 (5), I (12), II (40), III (22), IV (18)	DE
(MET)	CRC post-S	12	63.9 (SD: 12.5)	10/2	0 (0), I (4), II (4), III (2), IV (2)	
	CRC (6 m)	52	60.1 (SD: 11)	38/14	0 (0), I (12), II (17), III (15), IV (8)	
	CRC (12 m)	38	61.5 (SD: 11.6)	24/14	0 (0), I (7), II (13), III (14), IV (4)	
[45]	CRC	24	65.03 (SD: 10.43)	13/11	A (1), B (1), C (12), D (0)	CN
(MET)	Control	80	64 (SD: 9.87)	43/37	-	
[26]	Adenoma	15	66 (ND)	8/7	-	PL
(MET)	CRC	72	54 (ND)	31/41	ND	
	Control	56	65 (ND)	32/24	-	
[46]	CRC-Malignant	94	ND	-	ND	CN
(MET)	Control	34	ND	-	-	
[47]	Adenoma	10	ND	-	-	CN
(MET)	CRC	52	60 (R: 26–87)	29/23	A (7), B (23), C (15), D (7)	
	Control	60	52 (R: 21–71)	31/39	-	
[48]	Adenoma	80	67 (ND)ǂ	93/70ǂ	-	UK
(VOL)	CRC	12			ND	
	Control	83			-	
	Other (DD, Hemorrhoids, etc.)	33			-	
[49]	Adenoma	94	68 (R: 29–89) ^ǂ^	286/276 ^ǂ^	-	UK
(VOL)	High risk adenoma	27			-	
	CRC	35			ND	
	Control	233			-	
	Others (DD, IBD, MC, etc.,)	173			-	
[23]	Adenoma	155	59.9 (SD: 7.4)	95/60	ND	CA
(MET)	Control	530	56.1 (SD: 8.2)	222/308	-	
[27]	Adenoma	345	65.1 (SEM: 6.6)	197/148	ND	CN
(MET)	Control	316	61.8 (SEM: 7.4)	82/234	-	
* [50]	Adenoma	243	59.5 (SEM: 0.67)	145/98	ND	CA
(MET)	Control	633	55.8 (SEM: 0.47)	269/364	-	

ND not disclosed, R range, SD standard deviation, SE/SEM standard error (of the mean), EC esophageal cancer, LC lung cancer, GC gastric cancer, BrC breast cancer, BC bowel cancer, NpC nasopharyngeal cancer, LeukC leukemia cancer, LyC lymphoma cancer, IBS irritable bowel syndrome, S surgery, m month, DD diverticular disease, IBD inflammatory bowel disease, MC microscopic colitis, CAD Canadian recruitment, MSKCC Memorial Sloan Kettering Cancer Center (New York) recruitment, UK United Kingdom, CN China, KR South Korea, JP Japan, CA Canada, US United States, TW Taiwan, PT Portugal, DE Germany, PL Poland. ^ǂ^ Total of participants data. Cancer stages follow either T-stage (0, I, II, III, IV) or Dukes’ stage (A, B, C, D).

### 3.3. Systematic Review

The number of compounds identified in the 28 studies included was 244, with 81 compounds from the volatilome, 160 from the metabolome, and 3 classified in both. Each article reported the compound name translated to InChIKey with the chemical translation service (http://cts.fiehnlab.ucdavis.edu/batch (accessed on 14 September 2020), [51]). These results were compared to match the compound identifiers between articles, as not every author reported the same compound in the same manner. If a compound was not found by the CTS service, a manual search at PubChem (https://pubchem.ncbi.nlm.nih.gov/ (accessed on 14 September 2020)) was performed. In Appendix A, we provide a detailed list of all 244 compounds with their common names, MW, chemical formula, and major identifiers (InChIKey, PubChem ID, HMDB ID, KEGG ID, Canonical SMILES and CAS). Appendix A presents information on the behavior of the identified compounds in the studies from the systematic search. A repeated trend means that the compound was found in more than one comparison. The monoisotopic mass from all compounds ranged from 31.06 g/mol for methylamine with only 1 carbon to 365.50 g/mol for tetradecanoyl carnitine (C14:1) with 21 carbons. There were three compounds shared between the two phases (MET and VOL): phenol [29,35], p-cresol [35,41,43,44], and oxalic acid [28,44]. The compound most repeated in the literature was hippuric acid, identified by five studies from seven comparisons [24,35,36,37,43]. In the independent comparisons, we accounted for cohorts that were used for different purposes, for example, comparing colorectal cancer patients versus controls, as well as comparisons among colorectal cancer stages. The most repeated compounds in the literature after hippuric acid are citric acid [24,31,35,36,43], 1-methyladenosine [38,39,42,46,47], indole-3-acetic acid [24,35,37,44], and p-cresol [35,41,43,44]. In total, 244 metabolites related to colorectal cancer were identified; however, 174 compounds were reported just once. Otherwise, reliable identification of metabolites was not always achieved; 29 compounds (Appendix A) were reported in 3 studies [32,44,46] with only retention time (based on the methodology used) and could not be matched with a specific compound from libraries or standards. Compound identification was performed in 22 studies, and trends in compound levels were disclosed in 18 of them.

To evaluate the consistency among the results, we evaluated vote-counting and the number of identifications. Vote-counting consists of the sum of the trends reported for compounds, assigning a value of +1 if the compound behavior is upregulated, −1 if it is downregulated or 0 if it is equal to the comparison group. To find stronger identifications, we plotted the vote-counting results versus the total number of articles in which a compound was reported (see Figure 2 and Appendix A) for compounds reported in at least two articles. If the value of vote-counting diverges from the number of articles, then there is inconsistency in the results, as it means that the trends are not the same for all records from the articles evaluated. Any compound intended to be a CRC biomarker needs to be robust, meaning that it needs to be identified in more than one study, and these identifications need to all show the same trend. Compound grouping was not attempted during this step.

### 3.4. Quality Assurance

Quality assurance of the studies included in the systematic review was performed, including ten variables for evaluation. The quality assurance results are shown in Figure 3 and Appendix A. Variables were based on the experimental methodology. The most reported domains were in sample collection, sample preparation, and experimental conditions, with more than 50% of studies reporting complete information. On the other hand, the least reported domains were in study design and data preprocessing, where less than half of the studies disclosed some information.

### 3.5. Meta-Analysis

For the meta-analysis, we excluded those studies that did not fulfill the minimum requirement of *N* = 20 per group, matching groups by age and sex, compound identification, ethics approval reported, and urine storage at −80 °C. From the 28 articles on the systematic review, we retained 9 articles for the meta-analysis, 8 for the metabolome [24,25,35,36,37,43,44,50], and 1 for the volatilome [29]. The results were analyzed in two parts. First, vote-counting was performed, like the systematic review, for up to five different groups, as reported in the selected articles: (i) CRC and advanced adenoma vs. control; (ii) CRC stages vs. control; (iii) polyps vs. control; (iv) pre-surgery vs. post-surgery; and (v) CRC vs. breast and esophageal cancer. Comparing the same conditions between these new groups of compounds allows us to analyze the compounds more accurately. Second, a proper statistical analysis was attempted, but it was only possible for the CRC and advanced adenoma vs. control and pre-surgery vs. post-surgery groups due to the need for a fold-change and a significant number of articles. For the compounds reported in each group, pathway enrichment was performed with different databases and meta-searches [52,53,54,55].

#### 3.5.1. Vote-Counting Results by Group

For the 8 articles included in the meta-analysis, we analyzed them by experimental study classes. Table 3 shows a summary of the results of the compounds found in at least two different cohorts. In Appendix A, we detail all the results for the five different groups.

i. CRC and advanced adenoma vs. control

There were 96 compounds found to be significantly different between CRC patients or patients with advanced adenoma and healthy controls (see Appendix A). Of these compounds, 22 were reported in more than two different cohorts (see Table 3), but 11 of them were only reported by Deng et al. [24], who used two different cohorts, so we considered the results for each of the cohorts. The most repeated compounds were citric acid and hippuric acid; however, their behavior differed, leading to final count votes of −2 and −3, even though they were reported five and four times, respectively. We identified only six compounds with stable behavior: creatinine, phenol (downregulated), D-glucose, L-kynurenine, L-proline, and N1,N12-diacetylspermine (upregulated). The pathway enrichment analysis results for compounds with two or more equal records showed relations with arginine and proline metabolism and the urea cycle and the metabolism of amino groups. Phenol was the only compound shared between the volatilome and metabolome.

**Table 3 cancers-13-02534-t003:** Relevant compounds from the meta-analysis results considering only the vote-counting results. The compounds shown are found in at least two different cohorts. Compounds in bold have a vote count of at least 2 for the upregulated compounds and −2 for the downregulated compounds.

Common Name	No. of Cohorts	Behavior (Up-Down-Equal)	Vote-Counting	*N*	Reference
**CRC and Advanced Adenoma vs. Control**					
N1,N12-Diacetylspermine	3	3–0–0	3	928	[24,25] ^ɫ^
D-Glucose	2	2–0–0	2	696	[24] ^ɫ^
L-Kynurenine	2	2–0–0	2	696	[24] ^ɫ^
L-Proline	2	2–0–0	2	696	[24] ^ɫ^
Creatinine	2	0–2–0	−2	452	[35,36]
Phenol	2	0–2–0	−2	294	[29,35]
Putrescine	3	2–0–1	2	900	[24,35] ^ɫ^
Hippuric acid	4	0–3–1	−3	1148	[24,35,36] ^ɫ^
Indole-3-acetic acid	3	0–2–1	−2	900	[24,35] ^ɫ^
Citric acid	5	1–3–1	−2	1271	[24,35,36,43] ^ɫ^
P-Cresol	2	1–1–0	1	417	[35,43]
Tetradecenoyl carnitine (C14:1)	2	1–0–1	1	696	[24] ^ɫ^
2-Aminohexanedioic acid	2	0–1–1	−1	696	[24] ^ɫ^
3-(3-Hydroxyohenyl)-3-hydroxypropanoic acid	2	0–1–1	−1	696	[24] ^ɫ^
Aspartic acid	2	0–1–1	−1	696	[24] ^ɫ^
3-Hidroxybutyric acid	2	1–1–0	0	696	[24] ^ɫ^
Butyric acid	2	1–1–0	0	696	[24] ^ɫ^
Hydroxyproline	2	1–1–0	0	696	[24] ^ɫ^
L-Alanine	2	1–1–0	0	452	[35,36]
L-Dopa	2	1–1–0	0	696	[24] ^ɫ^
L-Tryptophan	2	1–1–0	0	327	[35,43]
Urea	2	1–1–0	0	452	[35,36]
**CRC Stage vs. Control**					
Hippuric acid	2	1–1–0	0	248	[37,44]
**Pre-surgery vs. Post-surgery**					
Salicyluric acid	2	0–2–0	−2	258	[43,44]
Asparagine	2	1–1–0	0	258	[43,44]
Citrate	2	1–1–0	0	258	[43,44]
Tyrosine	2	1–1–0	0	246	[43,44]

^ɫ^ Reference [24] has two cohorts for this is included twice.

ii. CRC stage vs. control

We found 24 compounds related to different disease stages (Appendix A) from the only two articles that studied different CRC stages included in the meta-analysis. The two included articles in the meta-analysis consider different groups of CRC stages for their analysis, consequently no proper CRC stages combination of results could be performed. The stages studied were stages I and II as a sole group versus controls [37], when tumors have not spread to the lymph nodes or other tissues (metastatic), and stages II, III, and IV (metastatic) combined as sole group versus early stages (0 and I) [44]. Regarding the behavior of the compounds, 13 were upregulated compared to controls, and 10 were downregulated. One compound was identified in both articles, hippuric acid, but conflicting results were obtained. There were no studies reporting compounds for the volatilome.

iii. Polyps vs. control

In the comparison of the metabolome of patients with polyps and controls, five compounds appeared, all of which were in the metabolome (Appendix A). Only one article studied compounds in patients with polyps against controls. Regarding behavior, most of the compounds presented the same levels between controls and patients with polyps. Only one compound, ethanol, was upregulated in patients with polyps, and four compounds were downregulated: 3-hydroxybutyric acid, 3-hydroxymandelic acid, adipose acid, and benzoate. Regarding the involved pathways, after enrichment analysis, significant results were obtained for protein digestion and absorption, central carbon metabolism in cancer, aminoacyl-tRNA biosynthesis, propanoate metabolism, and alcoholism. Because we considered at least two articles as the minimum for considering a compound consistent, we cannot draw firm conclusions about these compounds.

iv. Pre-surgery vs. post-surgery

When evaluating the differences in the metabolome in patients with CRC before surgery and after surgery, we found 46 compounds that were significantly differentiated from the selected studies included in our meta-analysis (Appendix A). Only compounds reported more than once were considered to ensure consistent results, and only four compounds met this criterion. An evaluation of trends showed that only salicyluric acid had the same trend in both studies and was downregulated when comparing urine samples before and after surgery from CRC patients, showing a reduction in compound levels after surgery. The other three compounds (asparagine, citric acid, and tyrosine), showed different trend behaviors between the two studies, so no firm conclusions could be drawn for them. Regarding the pathways involved, salicyluric acid is involved in biological oxidation and compound conjugation, such as amino acid conjugation. Asparagine, citric acid, and tyrosine are related to amino acid metabolism and significantly related to central carbon metabolism in cancer.

v. CRC vs. breast and esophageal cancer

In the comparison of the behavior of some compounds in the presence of different cancers, such as breast cancer and stage I/II esophageal cancer, 34 compounds were reported (Appendix A). Of these compounds, 21 are from the volatilome [29] and 13 are from the metabolome [37]; however, the VOLs are compared in breast cancer, while the METs are compared in esophageal cancer. Of the VOLs, 11 were downregulated and 10 were upregulated in breast cancer comparison; of the METs, 4 were downregulated and 9 were upregulated in esophageal cancer comparison.

vi. Comparisons among the different groups

When comparing all the different groups considered, at least 156 compounds were identified, as shown in Appendix A. Of those, 38 were found in at least two of the groups. The vote-counting of the comparison returned 24 compounds found at least twice (Appendix A). The most abundant compounds were N1,N12-diactylspermine (upregulated) and hippuric acid, creatinine and guaiacol (downregulated). Creatinine had the lowest vote count of −4, and it appeared downregulated in all the groups. N1,N12-Diactylspermine, hippuric acid, and guaiacol have vote counts of 3, −3, and −3, respectively. They had a similar trend in the groups where they were present, except for hippuric acid, which was downregulated in CRC and adenoma versus the control but also during the pre-/post-surgery period, while it was upregulated in CRC vs. other cancers and showed no significant difference across the CRC stage groups. No single compound was shared across all groups.

In Figure 4, we present the results for the compounds from (Appendix A) that have similar behavior and are reported in at least two groups, but excluding the CRC vs. other cancers groups, as we cannot ensure a similar trend as the case-controls. Additionally, the pre-/post-surgery group resembled the case-control group when considering cases as pre-surgery and controls as post-surgery. Finally, CRC stages were compared between early stages I/II and control or intermediate and late stages and early stages. In doing so, 10 compounds were found to be shared among the three groups. P-Cresol was upregulated in all the groups, except for the pre-/post-surgery. Hippuric acid was downregulated in all groups, except for the CRC stages, where it exhibited a different behavior.

#### 3.5.2. Statistical Results by Group

Statistical analysis was performed for those groups of studies that reported the p-values and fold-changes for significant compounds, and two groups remained for further study: (i) CRC and advanced adenomas with respect to controls and (ii) pre-surgery with respect to post-surgery. Only one article reported the concentration mean values and errors [43] of the significant compounds.

i. CRC and advanced adenomas vs. controls

The *p*-values and fold-changes reported in the articles (Appendix A) were combined to observe the global significance of the metabolites (Appendix A). Only included metabolite records with all information disclosed by the authors were included. The results are presented as a volcano plot in Figure 5, which shows a combined volcano plot for the upregulated compounds and downregulated compounds. Statistical significance (α ≤ 0.05 and fold-change > 4) was reached by ten metabolites (four upregulated and six downregulated), the largest fold-change reported was for butyraldehyde (downregulated), and the smallest *p*-value was for N1,N12-diacetylspermine. Only one of the statistically significant metabolites was relevant in the vote-counting analysis: hippuric acid. The combined and weighted *p*-values and fold-changes for the significant compounds are shown in Table 4, along with the normal urine concentrations of each compound in healthy individuals, as stated in the human metabolome database when available, and a projection of the range values for each compound in urine samples from a patient with colorectal cancer or advanced adenoma.

ii. Pre-surgery vs. Post-surgery

The p-values and fold-changes reported in the articles (Appendix A) were combined to observe the global significance of the metabolites (Appendix A). Only the metabolite records that had all information disclosed by the authors were included. The results are presented as a volcano plot in Appendix A, which shows a combined volcano plot for the upregulated and downregulated compounds. There were three compounds with statistical significance (α ≤ 0.05 and fold-change > 4): hippuric acid (upregulated), hydroquinone, and tartaric acid (downregulated). Salicyluric acid, the only compound reported in both articles, did not reach statistical significance.

iii. Comparisons among the different groups

Based on the comparison of CRC vs. controls and pre-surgery vs. post-surgery, 11 compounds were shared between the two groups. When CRC patients and pre-surgery patients were classified as CRC patients and controls and post-surgery patients were classified as controls (Appendix A), up to 12 compounds were shared between the two groups, and two compounds were repeated more than three times. Citric acid appeared in five articles, but it was downregulated in three of them and upregulated in two, and the results were not statistically significant after combining the studies (Appendix A). Hippuric acid and indole-3-acetic acid were downregulated in both groups, and 2-aminobutyrate and L-pyroglutamic acid were upregulated in both groups.

The combination of statistical information on both groups showed the same results as those in previous comparisons, and 11 compounds reached statistical significance (α ≤ 0.05 and fold-change > 4) (Appendix A). Four upregulated compounds were found, 3-hydroxybutyric acid, L-dopa, L-histidinol, and N1,N12-diacetylspermine, all of which were metabolites, and seven downregulated compounds were found, pyruvic acid, hydroquinone, tartaric acid, and hippuric acid as metabolites and butyraldehyde, ether, and 1,1,6-trimethyl-1,2-dihydronaphthalene, all of which were volatiles.

## 4. Discussion

One of the biggest efforts in this systematic review and meta-analysis was the combination of all relevant compounds from the selected articles. Each individual compound from each article was searched for its PubChem ID, and a compound name was selected if more than one was reported. However, this was not possible for all compounds; for example, the NMR technique allows us to discern between cis- and trans- of aconitic acid [37,44], which is not possible with regular GC or LC-MS techniques [43]. We have included several chemical identifications, so it will be easier to compare the results presented here with future results reported by the scientific community.

For the systematic review, 244 compounds were identified in urine samples among individuals with colorectal cancer or polyps in the studies, but only 68 were reported more than once. Of those, the most abundant compounds were organic acids and derivatives, comprising 66 compounds (including 40 amino acids and derivatives, 9 dicarboxylic and tricarboxylic acids and their derivatives, and 5 dipeptides), 30 benzenoids (8 phenols, 6 hippuric acids and derivatives, 3 naphthalenes), 22 lipids and lipid-like molecules (including 15 fatty acyls and 5 prenol lipids), and 20 nucleosides (including 14 purine, 5 pyrimidine nucleosides). Up to 163 compounds were metabolites (58 reported more than once), while only 84 were volatiles (13 reported more than once). Three of them were found to be both a metabolite and volatile (p-cresol, oxalic acid, and phenol).

The vote-counting results revealed seven upregulated compounds reported in at least three studies (1-methyladenosine, cytidine, pseudouridine, N6-methyladenosine, N1,N12-diacetylspermine, inosine, and adenosine) and three downregulated compounds reported in at least three studies (creatinine, indole-3-acetic acid, and hippuric acid). These 10 compounds, which are all metabolites, are involved in 16 pathways. Of the upregulated compounds, first, the 1-methyladenosine urine levels appeared elevated in patients with malignant tumors. Cytidine and pseudouridine are involved in nucleic acid synthesis. Inosine is involved in the degradation of purines for purine salvage, and its levels appear abnormal in urine in some metabolic and immune system disorders. Finally, adenosine is essential for life and is involved in many pathways, such as biological oxidation, signal transduction, and even energy transport, and its levels in urine appear altered in metabolic diseases and nervous system disorders [53,54].

Of the downregulated compounds, creatinine is involved in many pathways, such as vitamin, amino acid and polyamine metabolism, and the urea cycle, and abnormal levels have been reported in patients with metabolic disorders. Indole-3-acetic acid is also involved in amino acid pathways. In the case of hippuric acid, all results showed that it was downregulated in CRC; only when compared with another cancer (esophageal) did it seem to be upregulated. Hippuric acid is involved in metabolic pathways such as phenylalanine metabolism. On the other hand, there were 100 metabolites reported in a single study by Madhavan et al. [22] using UPLC-QTOF-MS, and 34 relevant metabolites were investigated to determine the differences between colorectal cancer patients with relapse and those without relapse. For polyp studies, the one that reported the most metabolites was by Wang et al. [50] using 1D NMR, who reported 17 relevant metabolites in polyp patients, but the trends of the comparison were not disclosed. As organic compounds, they are formed by carbon and hydrogen, but we can also see a high presence of nitrogen and, in smaller quantities, sulfur. In some studies, there is no complete identification of the compounds—only the molecular formula was disclosed, and it was not assigned to a compound, which makes comparisons more difficult. In three studies [24,27,50], the comparison trends were not disclosed, so we only know that these metabolites differ among groups but not if they are up- or downregulated.

For the meta-analysis, to ensure the quality of the results studied, we only included studies that fulfilled the conditions of minimum N, matched groups, those that identified the compounds and reported their behavior, those that provided ethics approval, and those that met the sample storage conditions. The results were divided into vote-counting results and statistical results by group. Not all articles reported the concentration mean and standard deviation, which are needed to create a forest plot (the plot usually used for a meta-analysis). In fact, metabolomics and volatilomics are burgeoning research fields that are currently expanding, which means that these scientific communities, even though there are some efforts in this respect, still lack a uniform manner of presenting their results. For that reason, we were only able to use the results of articles that provided p-value and fold-change information for the significant compounds to create a combined volcano plot of all the available results.

For the meta-analysis vote-counting results, in the case-control analysis, five compounds were upregulated (N1,N12-diacetylspermine, D-glucose, L-kynurenine, L-proline, and putrescine), and five were downregulated (creatinine, citric acid, indole-3-acetic acid, hippuric acid, and phenol).

Of the possible biomarkers obtained from the evaluation, only citric acid appeared in all papers that evaluated CRC patients versus healthy controls, but it did not show the same trend across the studies. Citric acid, or citrate, plays a role in metabolism in the citrate cycle (TCA), which forms part of carbohydrate and fatty acid oxidation (KEGG pathways: hsa00020). In cancer cells, de novo lipid synthesis supported by citrate is enhanced [56], ultimately leading to a decrease in citrate urinary excretion. In this case, our results are in accordance with the literature, but citrate does not have the specificity of biomarkers for colorectal cancer, as it is a general mechanism found in other cancer types. Recently, it was described in a review on amino acid metabolism, the TCA cycle and lipid metabolism in colorectal cancer cells as well as in early phases, such as adenoma. These alterations are not specific to CRC pathways, as has been described in other tumor types [57]. N1,N12-Diacetylspermine is the only compound identified three times with a consistent trend (upregulated), and some studies have evaluated this metabolite and its relation with colorectal cancer [58,59,60]. Creatinine is another metabolite identified more than once in the colorectal cancer group, but this metabolite is difficult to evaluate in urine samples. In many urine metabolomics protocols, a common practice is to normalize the volume of samples using the concentration of creatinine. However, due to this normalization it is not possible to observe significant results on creatinine. A similar process can be employed with urea, the most abundant metabolite in urine—in this case, with the utilization of urease enzymes to remove it from the samples. For the studies comparing colorectal cancer and advanced adenoma versus controls, any of these disclosed methodologies can be used, but this fact must be taken into account for further evaluation of results, as we see here that creatinine could be a possible biomarker for colorectal cancer. D-Glucose is an important metabolite in human metabolism, and it participates in numerous pathways, such as glycolysis and carbohydrate digestion and absorption. A relationship between glucose and colorectal cancer has been described, and high levels of glucose in blood are associated with a high risk of colorectal cancer [61,62], which can explain the increased levels of glucose in urine. No data related to comorbidities were provided, so we cannot assure that D-glucose in urine is not due to diabetes mellitus. L-Kynurenine was upregulated, and its role in metabolism is principally in L-tryptophan metabolism, which is involved in a variety of physiological functions related to the immune system, central nervous system, and intestinal microflora. New studies have suggested that in colorectal cancer, L-tryptophan is preferentially converted to L-kynurenine due to the upregulation of the MYC oncogene [63]. Another amino acid that was upregulated was L-proline, which is involved in amino acid metabolism and central carbon metabolism in cancer cells. Proline is used in hypoxic and glucose deprivation conditions by cancer cells, contributing to ATP synthesis [64,65], and our results in urine are in contrast with those from the literature. Hippuric acid, which is involved in phenylalanine metabolism, is a metabolite that can be increased by the diet. Even if hippuric acid can be detected in urine for colorectal cancer, among other diseases, the biological relation between the compound and metabolism in cancer is unclear [66]. Indole-3-acetic acid produced by gut microbiota in tryptophan metabolism has been reported in fecal samples from colon cancer patients [67]. Later, Loke et al. [68] reported that indole-3-acetic acid was only found in normal tissue samples, not colorectal samples, which may be in accordance with its downregulation in urine samples from colorectal cancer patients. Finally, putrescine upregulation is related to some amino acid metabolism and the biosynthesis of secondary metabolites; among others, putrescine in cancer acts as a growth factor in cancer cells [69]. The only compound found in both volatilome and metabolome in the colorectal cancer versus control comparison was phenol, which was downregulated. Phenol forms part of some metabolic pathways related to amino acids and proteins. However, the main role in cancer metabolism is neutralizing free radicals and modulating enzymes related to oxidative stress [70].

In the CRC stage analysis, only two studies [37,44] tried to identify different levels of metabolites between patients with early stages of CRC and the latest stages compared to controls. With this strategy, the aim was to find possible compounds that allow differentiation between colorectal cancer stages. Only hippuric acid was shared in both articles. The trends reported for this compound are conflicting, but group comparisons were not consistent across studies. In the case of Liesenfeld 2015 [44], the largest difference was found between intermediate stages and early stages (upregulated). On the other hand, in Wang 2017 [37], the largest difference was found between early stages and controls (downregulated). Although the comparisons were of different colorectal cancer stages, the results could not be combined for evaluation.

In the polyp analysis, only one study met the criteria to be included in the meta-analysis; thus, it was not possible to evaluate the consistency of the polyp versus the control comparison results.

In the pre-/post-surgery analysis, to determine whether the levels of disrupted metabolites caused by colorectal cancer were restored, two studies were included, but only four compounds were found in both studies, and the only compound consistently downregulated was salicyluric acid. Therefore, prior to surgery, patients had less excretion of salicyluric acid, which is the glycine conjugate of salicylic acid, a component of aspirin. Some studies have suggested that aspirin has a chemoprotective role in colorectal cancer, where salicyluric acid and its derivatives are metabolized by colon cells and exert a protective effect [71,72]. One compound found to be upregulated pre- and post-surgery, L-pyroglutamic acid, was also upregulated in CRC and advanced adenoma compared to controls.

In addition, some studies tried to identify the similarities and differences between different types of cancers, which was the case for two studies, each of them analyzing different phases: the gas phase [29] and the liquid phase [37]. The first one compared patients with breast cancer and CRC against controls, and the second one compared patients with stages I and II esophageal cancer. None of the 34 compounds identified was shared between the studies, possibly because of the phase differences.

When comparing the meta-analysis vote-counting shared between groups considering the three groups of case-control, CRC stages, and pre-/post-surgery, we found 17 compounds, 10 shared in more than one group. The most relevant compounds were N1,N12-diacetylspermine (upregulated) and creatinine, indole-3-acetic acid and hippuric acid (downregulated). Hippuric acid appears at abnormal levels in urine in conditions related to germ lines, the nervous system, and metabolic disorders, and it is related to metabolic pathways. Indole-3-acetic acid is involved in the tryptophan metabolic pathway and appears at abnormal levels in urine in conditions related to immune system and digestive system disorders. Two volatile compounds were found to be relevant, guaiacol and phenol, and the latter was also found to be a metabolite. The volatilome is formed by small compounds that are able to move to the gas phase; the metabolome is formed by all the small compounds in the liquid phase. The low number of shared compounds could be due to the limited number of studies on volatilomes, the chemical differences in compounds, and the differences in methodologies used.

Considering the vote-counting results in the meta-analysis, the pathway analysis results indicated that some of the pathways in the metabolism of amino acids and their derivatives are the most altered in colorectal cancer. These pathways include glutamate and glutamine metabolism, the interconversion of polyamines, alanine metabolism, and tyrosine catabolism. The other pathway found to be significantly altered was the biological oxidation of amine oxidase reactions. On the other hand, pathway analysis performed directly on the systematic review results showed that the most affected pathway was caffeine metabolism.

Finally, a meta-analysis was performed to obtain a more reliable list of compounds that are altered in colorectal cancer. To maintain the trends in compound behavior, we split data between up- and downregulated compared to controls, further reducing the estimations for each compound. Additionally, the statistical results were not reported in the same way as in all articles, and some did not even disclose all the information, so these studies were excluded. Data were reduced in such a way that, for some metabolites, only one evaluation was retained. Despite the reduction in metabolites, we performed a meta-analysis and obtained nine metabolites that were statistically significant for the case-control group. Only one metabolite, hippuric acid, was also significant in the vote-counting analysis (compounds reported in more than one cohort that were included in the meta-analysis). The reduction in vote counting is due to the fact that statistical information was not disclosed or the trends reported for the metabolite were conflicting. For fold-change, there were some difficulties; in most studies, it was not disclosed, or if it was disclosed, the results were on a logarithmic or linear scale. Additionally, the range of sensitivities of the techniques makes it complicated to evaluate all the values together. The majority of studies included used a quantification technique of the metabolites (NMR or targeted mass spectrometry), but a few only performed metabolite identification (untargeted analysis). In the meta-analysis, we included all technical approaches, even though different techniques might have different sensitivities, mainly due to the small amount of data disclosed and the reduction in the metabolites listed. However, the results were consistent across both analyses for two downregulated metabolites: N1-N12-diacetylspermine and hippuric acid.

The statistical meta-analysis for the pre-/post-surgery group determined that three downregulated compounds were relevant: hippuric acid, hydroquinone, and tartaric acid. Salicyluric acid, which was relevant in the vote-counting analysis, had a low p-value to be considered significant. Hydroquinone has a role in two pathways, ubiquinone and other terpenoid-quinone biosynthesis and in tyrosine metabolism. Similar to other metabolites, we see that hydroquinone is involved in amino acid metabolism. However, tartaric acid participates in glyoxylate and dicarboxylate metabolism, which is a part of carbohydrate metabolism.

The statistical meta-analysis results when considering pre-surgery as CRC and post-surgery as controls revealed eleven relevant compounds, nine of which had been already found in the case-control group (3-hydroxybutyric acid, L-dopa, L-histidinol, N1,N12-diacetylspermine, butyraldehyde, pyruvic acid, 1,1,6-trimethyl-1,2-dihydronaphthalene, ether, and hippuric acid) and two of which were found pre-surgery vs. post-surgery (hydroquinone and tartaric acid). Three compounds were volatiles, and nine were metabolites. A biochemical and chemical similarity network (Figure 6) was developed based on the combined and weighted *p*-values and fold-changes in the relevant metabolites identified through the volcano plot, with at least a ± 1.5 log2FC. MetaMapp [73] was used to create the network, and Cytoscape [74] was used for visualization purposes. The network showed that a few volatile compounds were related to other metabolites. Brighter colors in the figure indicated the compounds with a minimum of ± 2.0 log2FC. The network analysis showed relations among volatiles and metabolites that emerged from the recurrence of their relations in different pathways.

It is interesting to note that most volatiles were not in the main group, and the volatiles that were in the main aggregation are close to each other and interrelated (P-cresol, phenol and 2-bromo-4-tert-butylphenol). Hippuric acid and P-cresol were found in both groups (case-control and pre-/post-surgery). Hippuric acid seems to be a recurring compound in all the analyses that we performed. Goveia et al. [75] found that hippuric acid was the only compound with a *p*-value < 0.1 for urine after evaluating up to 12 kinds of cancers in 25 studies. However, they did not implement a weighted *p*-value or include an evaluation of the fold-changes. They did include the vote-counting approach, for which hippuric acid had a value of −9 from the 11 studies included in the *p*-value evaluation, but no disclosure of the included cancers was provided. Despite these limitations, the behavior we observed in our analysis in colorectal cancer was consistent with that reported by Goveia et al. Additionally, hippuric acid has been reported as an upregulated marker of fruit and vegetable intake [76]. However, hippuric acid is commonly altered in almost all malignancies and a wide variety of other diseases [77] and the urinary metabolite is most strongly related to fecal microbial richness [78]. The lack of specificity of hippuric acid means that caution must be exercised when translating these findings to clinical applications.

This systematic review and meta-analysis is the first one to combine both metabolomics and volatilomics biomarkers for colorectal cancer in urine. There have been other attempts to account for urinary metabolome and volatilome in colorectal cancer [79] however, they were performed as a narrative review which included also other kinds of markers like proteins and genes. With additional investigations being published by the scientific community, we envision that some of the relevant volatiles and metabolites found might be reaffirmed as relevant and others might become irrelevant for colorectal cancer diagnosis via urine samples. We also expect that further live meta-analyses will be performed, as it is the intention of the authors to do so with the work presented here.

## 5. Conclusions

There are many compounds that can be suggested as possible biomarkers for CRC both in metabolomics and volatilomics in urine. Regarding volatile candidates, a large number of volatiles have been proposed as possible biomarkers. In contrast, the possibility of finding these biomarkers in different studies is very low. This may be because of the low number of studies that performed metabolite identification or the use of different techniques for urine analysis. Additional studies should be carried out to identify possible biomarkers that could be shared with those presented in this article.

We encourage the volatilomics and metabolomics communities to fully disclose their ethics approvals and the storage conditions of their samples, to report compound names along with at least one identifier and to include both *p*-values and foldchanges for the relevant compounds. Additionally, if more studies report the mean values with their standard deviation or even the raw data, this will allow for further meta-analyses using a standard forest plot.

Many discrepancies were found between the studies, for example, in metabolite behavior. A small number of studies in each group correctly evaluated the results. Although the patients included in these studies are from across the globe, a multicentric study is necessary. Another point that should be considered is the analysis and study of the same cohort with techniques to account for both metabolomics and volatilomics compounds. Finally, the reproduction of some of these studies would enable the use of these metabolites as biomarkers and is highly desirable.

To date, this is the first attempt of a combined systematic review of volatilomics and metabolomics data, in which we were able to identify 244 compounds in urine samples related to colorectal cancer. This is also the first meta-analysis evaluating colorectal cancer changes in urine samples, for which we found up to 11 compounds that might be specific biomarkers of colorectal cancer.

## Figures and Tables

**Figure 1 cancers-13-02534-f001:**
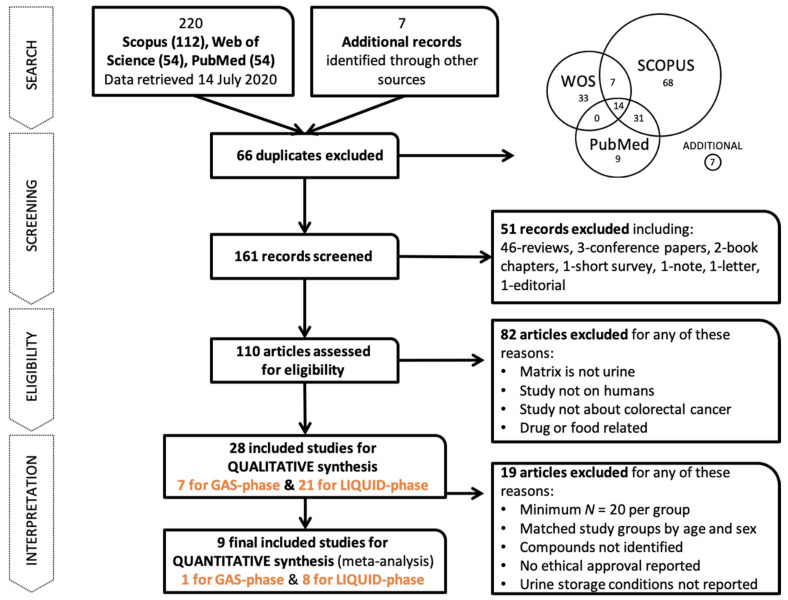
The whole workflow of the systematic review.

**Figure 2 cancers-13-02534-f002:**
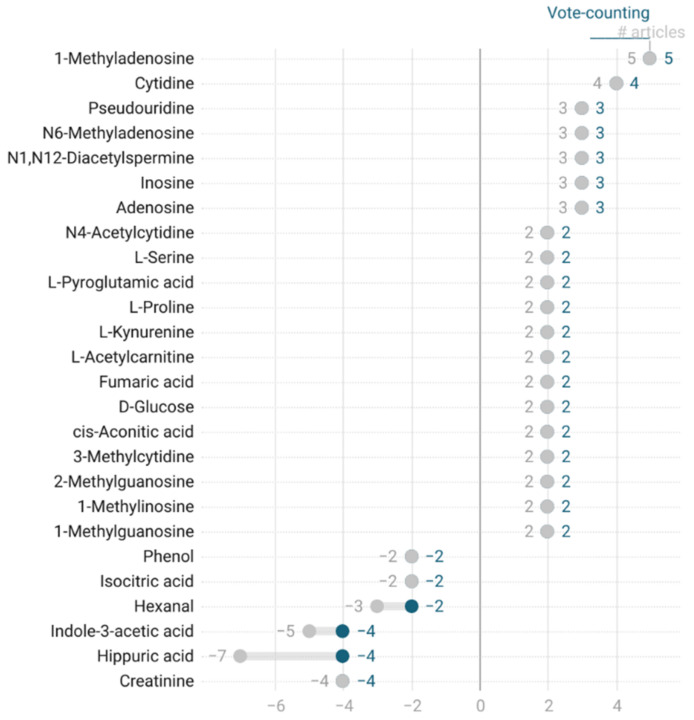
Qualitative vote-counting of colorectal cancer-related compounds, as a range plot between vote-counting values (blue) and total number (#) of articles (gray) from where the vote-counting is calculated. Positive values are compounds upregulated in CRC, while negative values are compounds downregulated in CRC.

**Figure 3 cancers-13-02534-f003:**
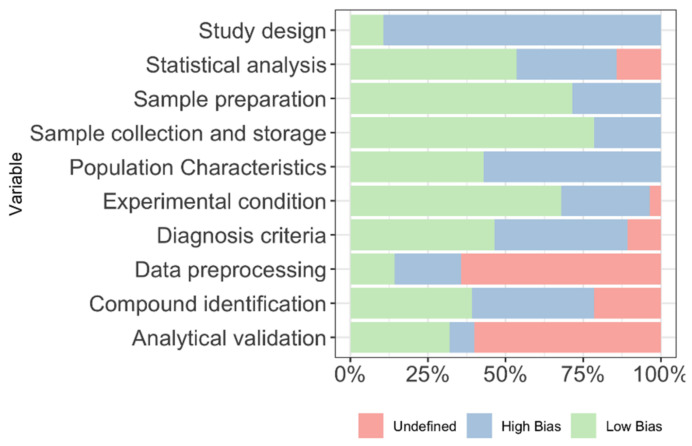
Quality assessment results for the included studies.

**Figure 4 cancers-13-02534-f004:**
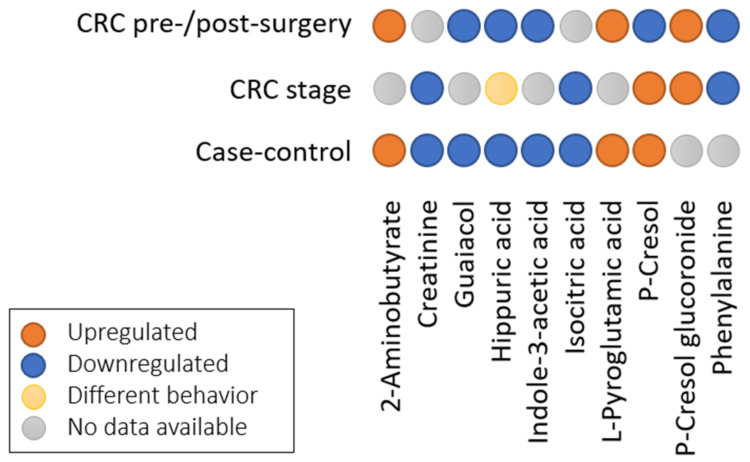
Comparison of shared compounds among three of the groups in the meta-analysis with vote-counting.

**Figure 5 cancers-13-02534-f005:**
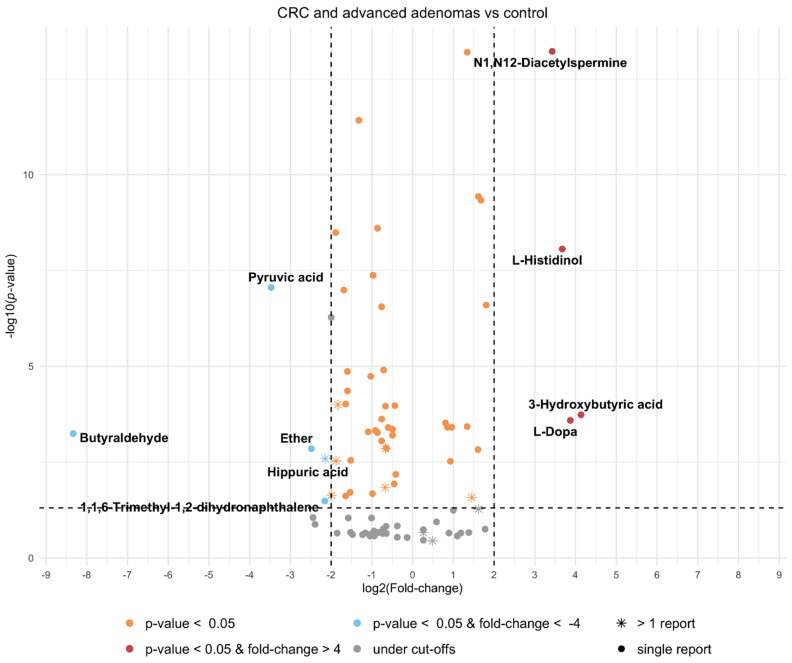
Volcano plot of compounds with consistent classification from meta-analysis on colorectal cancer and advanced adenomas vs. controls comparison. ⁕ Indicates the compound was found in more than one study.

**Figure 6 cancers-13-02534-f006:**
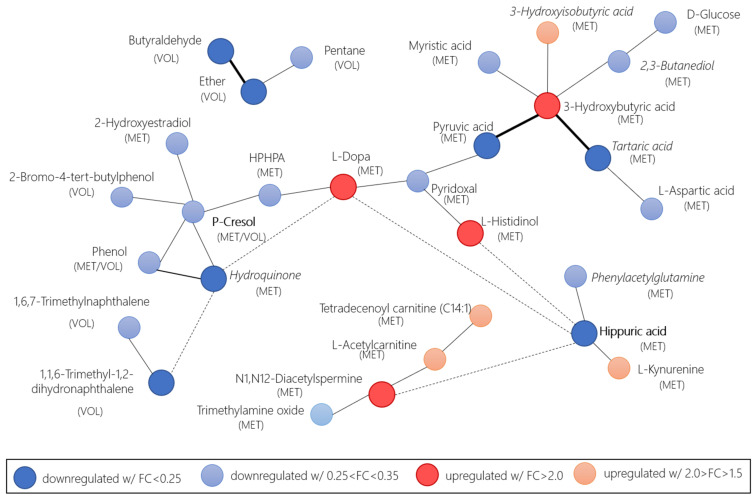
Network of relevant compounds in the comparison of CRC and adenomas versus control and CRC pre- and post-surgery, calculated by the weighted p-value and fold-change with MetaMapp biochemical mapping. Red nodes of upregulated compounds with FC > 2.0 (dark red) and 2.0 > FC > 1.5 (light red). Blue nodes downregulated with FC < 0.25 (dark blue) and 0.25 < FC < 0.35 (light blue). Thicker edges indicate that the relevant compounds are found in the volcano plot with log2FC above/below +2.0/−2.0. Dashed edges indicate that they are only reported for the log2FC above/below +2.0/−2.0 compounds. Bold names indicate compounds found in both the case-control and pre-/post-surgery groups. Compounds in the cursive were only found in the pre-/post-surgery group. HPHPA is the 3-(3-Hydroxyphenyl)-3-hydroxypropanoic acid.

**Table 4 cancers-13-02534-t004:** Meta-analysis results for the significant compounds in colorectal cancer and advanced adenoma vs. control comparison, including the healthy normal urine concentrations (extracted from the human metabolome database) and the projected colorectal cancer concentrations in urine. NQ: not quantified in urine.

Compound Name	Combined and Weighted *p*-Value	Combined and Weighted Fold-Change	*N* Total	HMDB ID	Healthy Normal Urine Concentration (Adult > 18 y) (µmol/mmol Creatinine)	CRC Projected Urine Concentration (Adult > 18 y) (µmol/mmol Creatinine)
3-Hidroxybutyric acid	1.85 × 10^−4^	17.56	342	HMDB0000357	1.4–2.7	5.8–11.2
L-Dopa	2.60 × 10^−4^	14.63	342	HMDB0000181	0.01–0.04	0.04–0.15
L-Histidinol	8.71 × 10^−9^	12.76	204	HMDB0003431	NQ	-
N1,N12-Diacetylspermine	6.00 × 10^−14^	10.75	342	HMDB0002172	0–0.0260	0–0.280
1,1,6-Trimethyl-1,2-dihydronaphthalene	3.31 × 10^−2^	0.22	60	HMDB0040284	NQ	-
Hippuric acid	2.59 × 10^−3^	0.23	546	HMDB0000714	28–610	6–140
Ether	1.42 × 10^−3^	0.18	60	-	NQ	-
Pyruvic acid	8.82 × 10^−8^	0.09	204	HMDB0000243	1–3.7	0.09–0.33
Butyraldehyde	5.76 × 10^−4^	0.003	60	HMDB0003543	NQ	-

## Data Availability

The data presented in this study are openly available in Zenodo at [10.5281/zenodo.4681360].

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
