# Peer review of "Comprehensive Volatilome and Metabolome Signatures of Colorectal Cancer in Urine: A Systematic Review and Meta-Analysis"

_cancers, 2021, doi:10.3390/cancers13112534_

Round 1
Reviewer 1 Report
An article devoted to the analysis of studies of the metabolome and volatile compounds of urine in colorectal cancer in order to search for putative markers of this type of cancer is presented for review. Review were performed using the Preferred Reporting Items for Systematic Reviews and Meta-Analyses or PRISMA method. The reliability of the data presented is beyond doubt. The review is well structured, accompanied by well-chosen illustrative material. An undoubted advantage is not a simple listing of the studies known in this field, but the analysis and systematization of the results, as a result of which 11 compounds were identified that could potentially be used for the diagnosis of colorectal cancer. I believe that the article can be recommended for publication in its present form. A small note. On line 163-164 there is no reference "Error! Reference source not found."
Author Response
Thanks for considering our article for publication in Cancers. We are glad that you find the article interesting and suitable for publication in its present form.
Regarding your small note. On line 163-164 there is no reference "Error! Reference source not found." It was an internal error in which this reference was a link to the Figure 1. We have already addressed this mistake in the manuscript by removing the link.
Thanks again for your comments and time reviewing our systematic review and meta-analysis. You will find enclosed the revised manuscript along with the extra information supporting some of the reviewers questions.

Reviewer 2 Report
The Review article authored by Dr Mallafré et al aims to summarize the current knowledge of the volatilome and metabolome of colorectal cancers in urine. This report encompasses a systemic review and a meta-analysis of the literature concerning this topic. The manuscript is well written and pleasant to read.
The strategies of data collection, selection/ exclusion criteria and mining for the systemic review and the meta-analysis are clearly described. The report covers many aspects on the knowledge available on urine volatilome and metabolome and their potential use as biomarker for the diagnosis of colorectal cancers, including selectivity towards tumors grade, expression in pre vs post surgery, links with metabolic pathways, specificity towards the type of cancer.
The meta-analysis revealed that 4 up-regulated and 7 down-regulated metabolites were reproducibly identified as potential biomarkers. The report also highlights the limits of some available data (ethical consideration, urine collection, sample conservation, experimental approaches, absence of compound identification, lack of statistical analysis) and raises some guidelines for the future studies in order to improve and normalize the analyses.
This topic is of importance, since the definition of urine biomarkers for colorectal carcinogenesis enables a noninvasive approach for early diagnose of cancer that should raise less aversion than fecal immunological testing –also associated with limited positive predictive value-. This manuscript should therefore make a suitable contribution for Cancers.
This exhaustive manuscript raises few comments
Does tumor size impact the metabolite accumulation in urine and thus tumor detection ?
The identification of an universal biomarker for colorectal cancer is very attractive. Nevertheless, colorectal cancers are no more considered as a uniform disease entity (c.f. the consensus molecular signatures, CMS1-4). As mentioned in the manuscript, c-MYC overexpression might be associated with L-kinurenine urine accumulation. Does an investigation of metabolomic/volatilomic biomarkers taking into account retrospectively these different subgroups -by decreasing the loss of information due to gathering data from different types of colorectal cancers- might allow the definition of more selective signatures and improve the sensitivity of detection ?
Few misprints need to be fixed, e.g.
Line 174 according to Figure 1, 8 studies were included for quantitative synthesis
Line 544
Line 562 “depravation”
Author Response
Thanks for considering our article for publication in Cancers. We are glad that you find the article interesting and pleasant to read. As well, we appreciate the detail with which you have reviewed our article in order to help us to correct the misprints.
Regarding the comments you raised:
- Does tumor size impact the metabolite accumulation in urine and thus tumor detection ?
In theory yes, size should have an impact on the metabolite accumulation and therefore, detection. However, with the data we collected from the systematic review, no specific meta-analysis could be done for CRC stages. From the initial 11 articles in the systematic review that reported the CRC stages, up to 8 were included in the meta-analysis. However, from those, only 2 had available or relevant data on the stages CRC. You can find more information in the table enclosed.
- The identification of an universal biomarker for colorectal cancer is very attractive. Nevertheless, colorectal cancers are no more considered as a uniform disease entity (c.f. the consensus molecular signatures, CMS1-4). As mentioned in the manuscript, c-MYC overexpression might be associated with L-kinurenine urine accumulation. Does an investigation of metabolomic/volatilomic biomarkers taking into account retrospectively these different subgroups -by decreasing the loss of information due to gathering data from different types of colorectal cancers- might allow the definition of more selective signatures and improve the sensitivity of detection?
In theory if we could have enough data of CRC in urine with different molecular signatures from different articles, and if all of them pass the meta-analysis selection, yes, it could be done. Regarding the sensitivity of detection, it is specific to each experimental technique used and might be related to the chemical extraction done (in the case of LC-MS). For NMR instrumentation, the results can be improved, for instance, by performing multiple scans and using a higher magnetic field strength. The kind of molecule can also play an important role in sensitivity as some had better ionization patterns than others (LC-MS) and they derivatize better or not (liquid injection GC-MS). Also, newer equipment is being developed by metabolomics vendors that are decreasing the sensitivity of detection. And for volatilomics, newer methods are being developed, like the SPME arrows which are thicker and there for have more absorbent material.
- Misprint Line 174 according to Figure 1, 8 studies were included for quantitative synthesis,
Many thanks for the appreciation, we got confused with the numbers. There are 9 articles included for meta-analysis, as it was not correctly indicated we have corrected figure 1 and text line 246 indicating it.
- Misprint Line 544.
We absolutely agree with it, the sentence on line 544 was confusing. We have tried to address it reformulating the last part of the sentence to give more clarity.
BEFORE: In many metabolomics protocols for urine, a common practice is to normalize the volume of samples using the concentration of creatinine because normalization is possible not observe significant results on creatinine.
NOW: In many urine metabolomics protocols, a common practice is to normalize the volume of samples using the concentration of creatinine. However, due to this normalization it is not possible to observe significant results on creatinine.
- Misprint Line 562 “depravation”
This has been addressed by changing “depravation” to “deprivation”. Here we had a typo, as we are referring to the oxygen privation on that sentence.
Thanks again for your comments and time reviewing our systematic review and meta-analysis. We have tried to answer the best that we could all your concerns and made the proper changes to the manuscript. You will find enclosed the revised manuscript along with the extra information supporting some of the reviewers questions.

Reviewer 3 Report
Dear author, I have read with interest this manuscript which concerns an overall interesting topic. This paper is original and pave the way for further research in the field of volatilomes and metabolomes in colorectal cancer patients.
Please have the paper professionally proofread.
I would ask to the author to clarify when their findings were associated to a specific tumor stage or not. It might be interesting to find, if present, differences in terms of staging such as M0 versus M1. Were those patients with metastatic colorectal cancer included or not? Can you make separate findings?
Author Response
Thanks for considering our article for publication in Cancers. We are glad that you find the article interesting.
Regarding the comments you raised:
- Please have the paper professionally proofread.
We are surprised that you found the English article not good enough. We had the systematic review and meta-analysis article professionally proofread by the American Journal Experts before submitting it to Cancers. You can find their summary of the corrections enclosed.
- I would ask to the author to clarify when their findings were associated to a specific tumor stage or not. It might be interesting to find, if present, differences in terms of staging such as M0 versus M1. Were those patients with metastatic colorectal cancer included or not? Can you make separate findings?
A clarification has been included (revised manuscript, lines 377-379): “The two included articles in the meta-analysis consider different CRC stages, consequently no proper CRC stages specific combination of results could be performed.”
In theory tumor size should have an impact on the metabolite accumulation in urine and therefore, detection. However, with the data we collected from the systematic review, no specific meta-analysis could be done for CRC stages. From the initial 11 articles in the systematic review that reported the CRC stages, up to 8 were included in the meta-analysis. However, from those, only 2 had available or relevant data on the stages CRC. You can find more information in the table enclosed.
For the colorectal cancer studies included, one does early stages (stages I and II) in respect to controls, and the other one does intermediate (II and III) and late stages (IV, metastatic) in respect to early stages (0 and I). Thanks to your comment we have revised the information included in Table S7. Now it has been changed the following entrances:
“CRC_late-and-intermediate in respect to early stages” TO “CRC - stage II/III/IV vs 0/I”.
“CRC_late in respect to early stages (NMR values)” TO “CRC - stage IV (metastatic) vs 0/I and II/III (NMR values)”
We have clarified this in the text, and now can be read: “The stages studied were stages I and II as a sole group versus controls [37], when tumors have not spread to lymph nodes or other tissues (metastatic), and early (0 and I), intermediate (II and III) and late (IV, metastatic) [44].”
Thanks again for your comments and time reviewing our systematic review and meta-analysis. We have tried to answer the best that we could all your concerns and made the proper changes to the manuscript. You will find enclosed the revised manuscript along with the extra information supporting some of the reviewers questions.
